# Next-visit prediction and prevention of hypertension using large-scale routine health checkup data

Chung-Che Wang[1], Ta-Wei Chu[2,3]*, Jyh-Shing Roger Jang[1]

**1** Department of Computer Science and Information Engineering, National Taiwan University, Taipei, Taiwan, **2** Department of Obstetrics and Gynecology, Tri-Service General Hospital, National Defense Medical Center, Taipei, Taiwan, **3** MJ Health Screening Center, Taipei, Taiwan

* taweichu@gmail.com

**Data Availability Statement:** Data cannot be shared publicly because the copyright is held by the MJ Health Research Foundation. For researchers who tend to access the data, please

## Abstract

This paper proposes the use of machine learning models to predict one's risk of having hypertension in the future using their routine health checkup data of their current and past visits to a health checkup center. The large-scale and high-dimensional dataset used in this study comes from MJ Health Research Foundation in Taiwan. The training data for models is separated into 5 folds and used to train 5 models in a 5-fold cross validation manner. While predicting the results for the test set, the voted result of 5 models is used as the final prediction. Experimental results show that our models achieve 69.59% of precision, 77.90% of recall, and 73.51% of F1-score, which outperforms a baseline using only the blood pressure of visitors' last visits. Experiments also show that a visitor who performs a health checkup more often can be predicted better, and models trained with selected important factors achieve better results than those trained with Framingham risk score. We also demonstrate the possibility of using our models to suggest visitors for weight control by adding virtual visits that assume their body weight can be reduced in the near future to model input. Experimental results show that around 5.48% of the people who are with high Body Mass Index of the true positive cases are rejudged as negative, and a rising trend appears when adding more virtual visits, which may be used to suggest visitors that controlling their body weight for a longer time lead to lower probability of having hypertension in the future.

## Introduction

Hypertension is a health problem which increases the risk of many other diseases and one of the major cause of death [1, 2]. Therefore, early prediction and prevention while performing routine health checkup is an important issue.

Some research might focus on current-visit prediction of hypertension, where the extraction of input features and output are at the same visit to a health checkup center or hospital. AlKaabi et al. [3] used noninvasive factors, including age, smoking habit, and medical history, and employed decision tree, random forest [4], and logistic regression algorithms to predict

contact the MJ Health Research Foundation via e-mail: contact_us@mjhrf.org.

**Funding:** The author(s) received no specific funding for this work.

**Competing interests:** The authors have declared that no competing interests exist.

hypertension. Zhao et al. [5] did a similar task using a larger dataset and added different models including CatBoost, neural network, and logistic regression for the prediction. Despite that the model accuracy is high, the systems may not be helpful in practical use since measuring blood pressure can be convenient.

The next-visit prediction is possibly a more challenging task. Kanegae et al. [6] predicted new-onset hypertension at the third year by the features extracted in the first and the second year using XGBoost [7], logistic regression and a combination of them. Fang et al. [8] invoked several methods including LightGBM [9] and K-nearest neighbors for predicting the risk of hypertension within the next five years. Despite that the performance reported in [6, 8] are high, some features like cardio-ankle vascular index, thrombin time, and international normalized ratio are not included in a routine health checkup process. Besides, the input time range of the above studies are fixed at one or two years, which are not able to utilize the data of people who have routine health check-ups for several years.

In this paper, we use XGBoost, LightGBM, and random forest to predict the risk of hypertension for a visitor to a health check-up center at their next visit using their routine health checkup data, since these models achieved high performance in previous work. Similar to [3, 8], our experiments are conducted in a 5-fold cross validation manner. Besides, we also try to use our model to give suggestions of blood pressure controlling to visitors who are with high probability of having hypertension in their next visits. This paper is based on our previous work [10] (which was peer-reviewed and presented in a conference without a published full-text), and more experiments, detailed descriptions, and analysis are conducted in this paper. Detailed comparison of data and methods of our work and previous research is listed in Table 1.

## Materials and methods

This Section presents the ethical statement and a brief description of the data used in this paper, followed by the methods of data processing, model training, and utilizing trained models for hypertension prevention. An overview of the data processing and model training methods is illustrated in Fig 1.

### Dataset

This research, conducted under the oversight of the "NTU Behavioral and Social Sciences Research Ethics Committee" (Approval Number: 202002HM009), ensured the complete

**Table 1. Summary of prior research and our work.**

| Reference | Population | Number of visits for input | Interval between input and output | Is next-visit prediction | Methods |
|---|---|---|---|---|---|
| AlKaabi et al. [3] | 987 | 1 | None | No | Decision tree, random forest*, and logistics regression using 5-fold cross-validation |
| Zhao et al. [5] | 29,700 | 1 | None | No | Random forest*, CatBoost, MLP neural network and logistic regression using 10-fold cross-validation |
| Kanegae et al. [6] | 18,258 | 2 | 1 year | Yes | XGBoost* and logistic regression |
| Fang et al. [8] | 33,255 | 1 | 1 to 5 years | Yes | KNN and LightGBM* using 5-fold cross-validation |
| Ours | 74,802 | At most 5 | More than 1 year in most of the cases | Yes | XGBoost, LightGBM, and random forest using 5-fold cross-validation |

Models with the best F1-scores in each previous work are marked with asterisks.

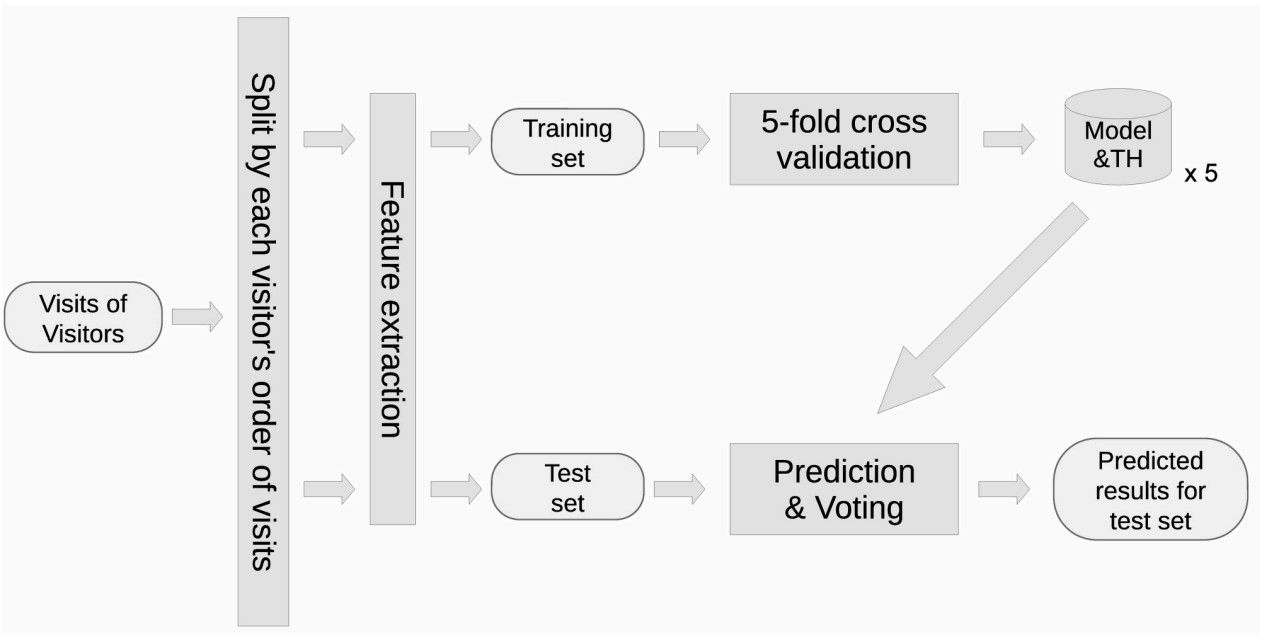

**Fig 1. An overview of methods of data processing and model training.** TH: threshold for binary decision.

anonymity of all data. The original health checkup data is held by the MJ Health Research Foundation, and the details of data collection methods have been reported in previous studies [11, 12]. This study utilizes secondary data from January 2010 to December 2017 and does not involve the recruitment of participants. All participants provided informed written consent. For minors, consent was obtained from their parents or guardians. The data used in this study was accessed on December 25, 2021, and all data used in this study is anonymous. Participants' confidentiality was rigorously protected throughout the study, adhering to all ethical guidelines and regulations governing human research. After removing visits of which blood pressure records are missing, there are 207,488 unique visitors, including 101,188 males and 106,300 females. The total number of visits is 382,610, where the distribution of number of visits for visitors is shown in Fig 2, and the distribution of intervals of every two neighboring visits is shown in Fig 3.

For visits of a single visitor, we first exclude a visit' all prior visits if the interval between the visit and its prior one is more than three years. For the remaining $N + 2$ visits, the first $N + 1$ visits ($V_1$ to $V_{N+1}$) are used for constructing the training set. Each training sample is composed of at most $N_1$ consecutive visits for input and the next one visit for output. At most last $N_2$ input-output pairs are considered for a single visitor. Similarly, the last $N_1 + 1$ visits are used for constructing the test set, also with at most $N_1$ consecutive visits for input and the next one visit for output. Note that if a visitor visits only twice, the corresponding data can only be used for the test set because there are no visits that can be used for the input of the training data; and visitors' data will not be used if they only visit once since no future data are available for comparing the groundtruth with the model prediction. We illustrate the above process of constructing training set and test set from one visitor's visits in Fig 4. In our experiments, we set $N_1$ to 5 and $N_2$ to 4, resulting in 40,488 training samples and 74,802 test samples from all visitors. The distribution of the number of visits for input for training and test sets is shown in Table 2.

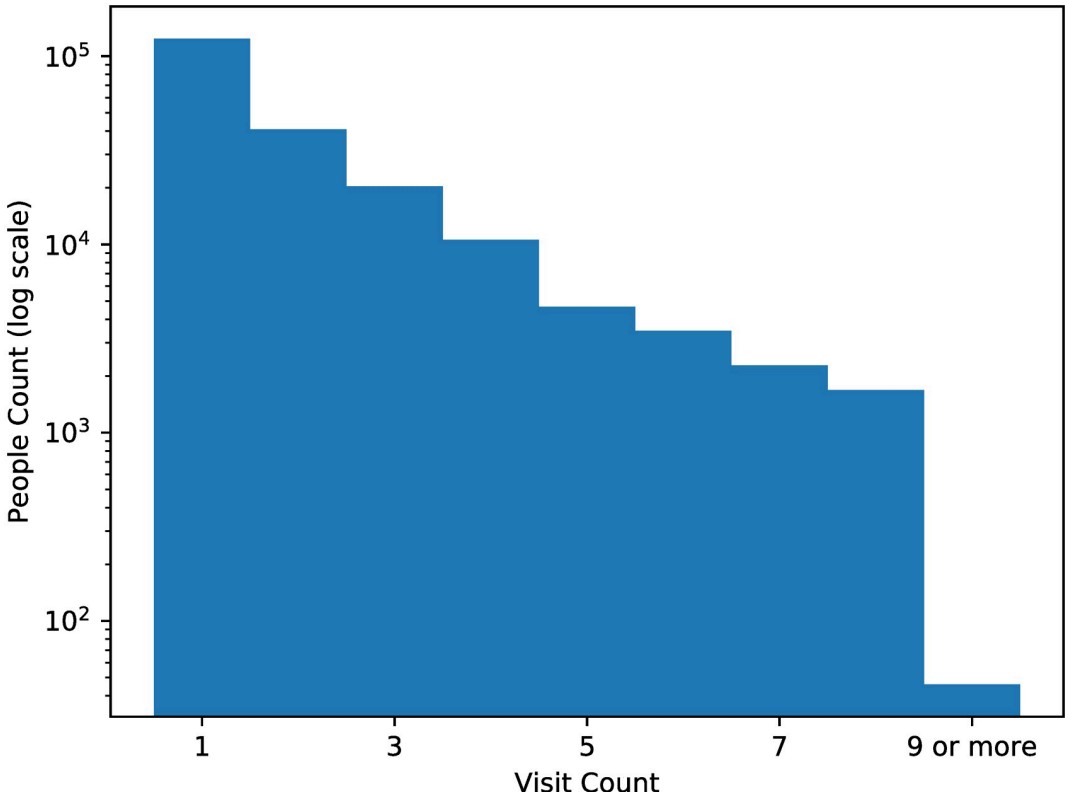

**Fig 2. Distribution of number of visits between January 2010 and December 2016 for all 207,488 visitors.**

### Feature extraction and output definition

For each visitor's each visit for input, we extract features from both physical examination results and questionnaire responses. The total number of feature dimensions $d$ for one visit is 266. A comprehensive list of the 266 features is shown in S1 Appendix. We will also list the important ones selected by our models in the experimental section.

For each visitor's all visits for input, we first concatenate the feature vectors of each visit according to their order of visiting. Second, to make the concatenated features the same size, we prepend zero vectors to the concatenated feature vector to length $N_1 \times d$ if the original size of the concatenated feature is less than $N_1 \times d$. Finally, to indicate if each of the $N_1$ sub vectors are prepended, we add a vector of size $N_1$ to the prepended vector. This step makes the models know if a sub vector is prepended by examining only one value. The final size of the feature vector is $N_1 \times d + N_1$, which is 1,335 in our experiments. Although using the above concatenation process requires a bit of extra storage and computational resources when training models, it utilizes data more effectively since data from visitors with different visit counts can be used together during training. Fig 5 shows an example of the above process of feature vector concatenation, which has three visits for input.

The process of handling missing values for one visitor's all visits for input is different for physical examination results and questionnaire responses. For physical examination results, we generally impute missing values for a visitor by linear interpolation based on the non-missing values, and keep them as missing if all values are missing. For questionnaire responses, we simply set a default response for each question, and use the default response if the

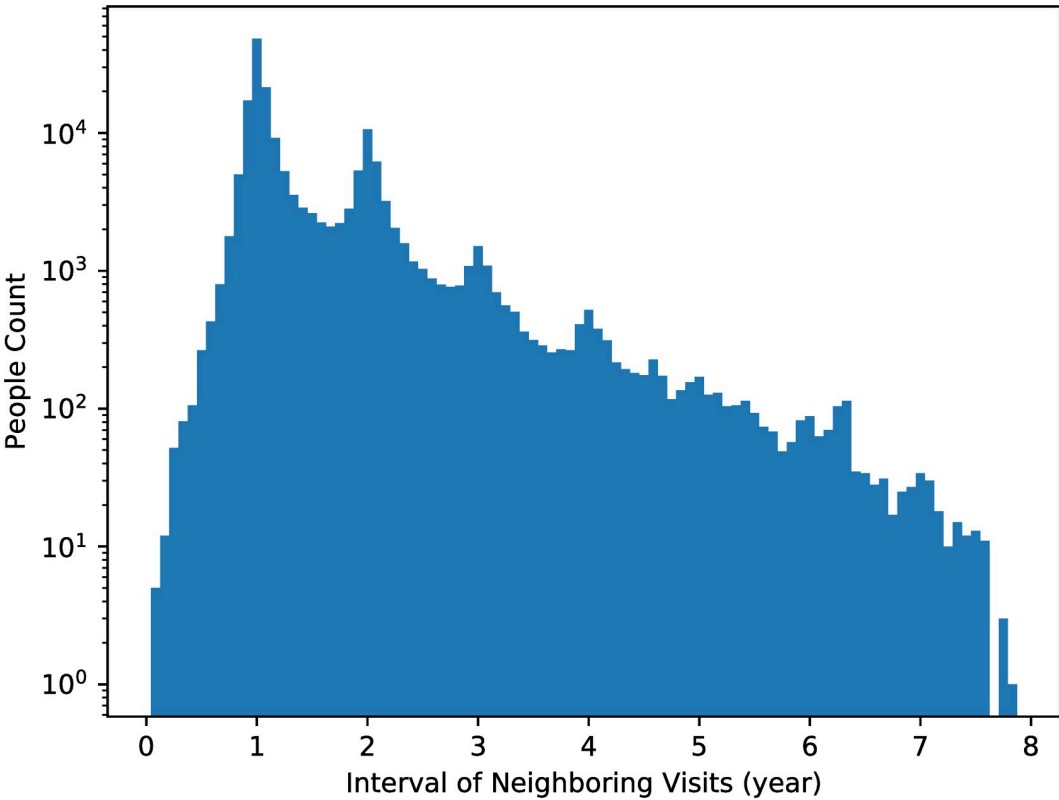

**Fig 3. Distribution of intervals of every two neighboring visits.**

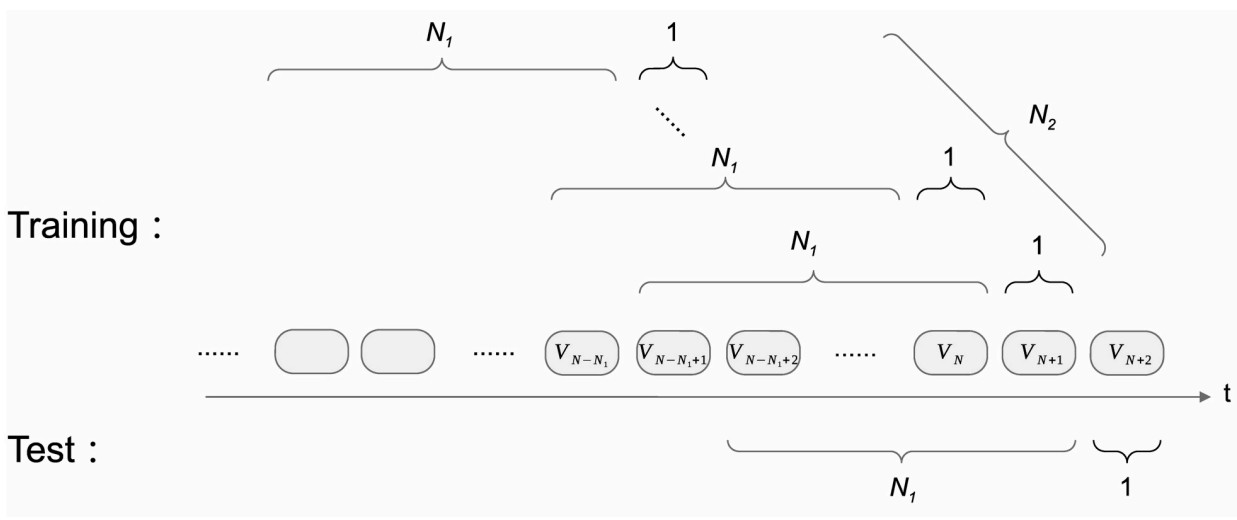

**Fig 4. The process of constructing training set and test set from one visitor's visits.**

**Table 2. Distribution of the number of visits for input for training and test sets.**

| Number of visits for input | 1 | 2 | 3 | 4 | 5 | Overall |
|---|---|---|---|---|---|---|
| Training | 17,587 | 9,458 | 4,286 | 3,387 | 5,770 | 40,488 |
| Test | 36,111 | 17,577 | 9,457 | 4,286 | 7,371 | 74,802 |

corresponding question is not answered. For instance, "No" is used as the default response of "Did you smoke?". A visit is defined as positive (i.e. having hypertension) if its systolic blood pressure is larger than or equal to 130 mmHg, or its diastolic blood pressure is larger than or equal to 80 mmHg. The ratio of positive visits for output is 34.03% for the training set and 33.63% for the test set.

## Train the models

This paper investigates the performance of random forest [4], XGBoost [7], and LightGBM [9], as these methods have achieved the best performance in various studies [5, 6, 8]. Random forest, XGBoost, and LightGBM are all tree-based methods, in which observations or features are randomly selected to build several decision trees using different algorithms, and the decisions of each tree are aggregated to obtain a final decision of a trained model.

Similar to [3, 8], the whole training set is separated into 5 folds, which are used to train 5 random forest, XGBoost, or LightGBM models in a 5-fold cross validation manner. To separate the training set, the whole set is firstly sorted using outputs' raw diastolic blood pressure as the primary order, and outputs' raw systolic blood pressure as the secondary order. The sorted set is then separated by the remainder of their indexes dividing by 5. In each fold, the model which achieves the best validation loss is used for this fold, and the probability threshold which achieves best F1-score for the validation set is used determine the output result of this model while evaluating using the test set. In the evaluation stage, the voted result of the 5 folds is used as the final prediction. Fig 6 illustrates the above process.

## Use of models for hypertension prevention

The most common suggestion of preventing hypertension or blood pressure control might be weight control [13, 14], and the scenario of using machine learning models to suggest one for

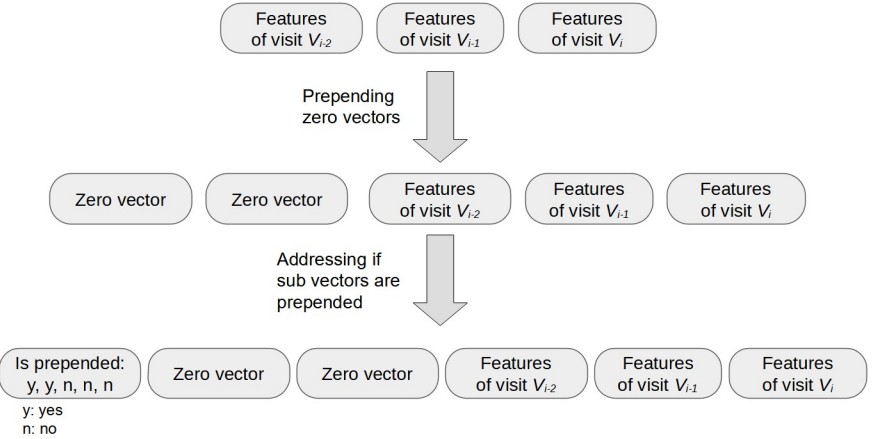

**Fig 5. An example of feature vector concatenation using three visits for input.** $V_i$ is the last visit for input.

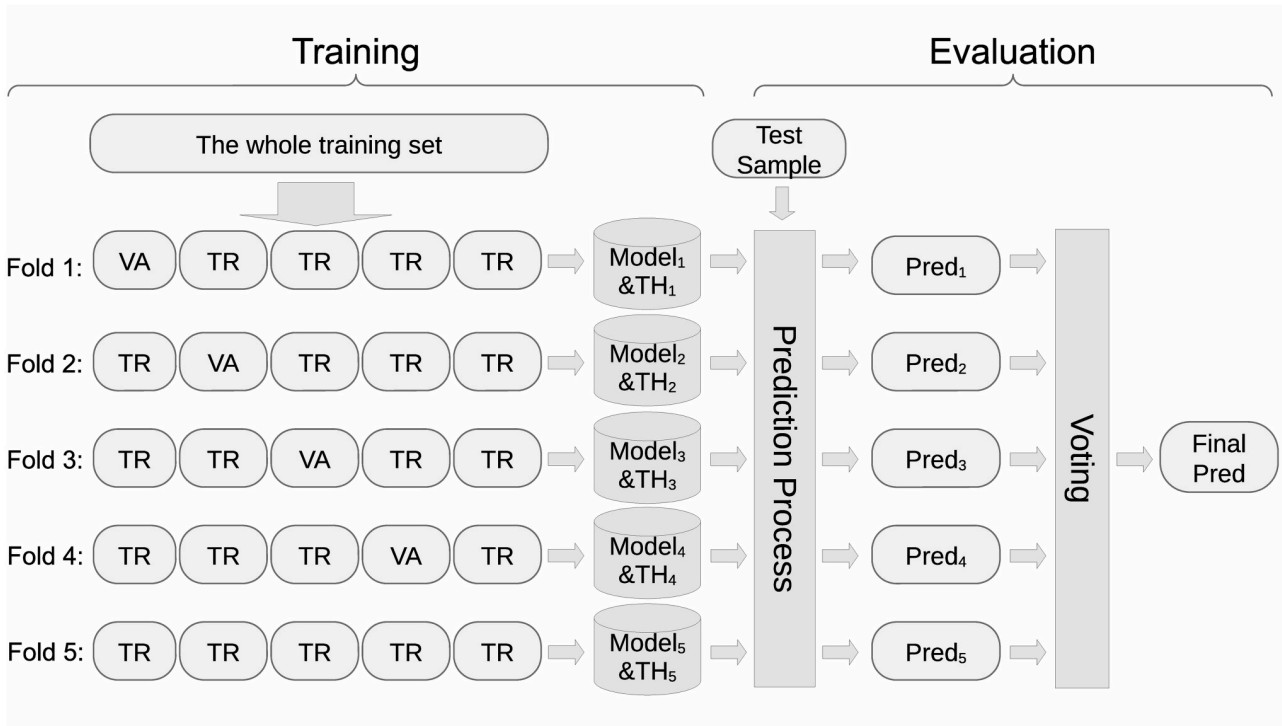

**Fig 6. Our process of 5 fold cross validation.** TR: separated subset used for training. VA: separated subset used for validation. TH: threshold determined by the VA of the corresponding fold. Pred: prediction.

hypertension prevention could be telling them to control their body weight in the near future, and after that they are expected to be with lower probability of having hypertension. To achieve that, for one visitor who is predicted to have hypertension on their next visit, we first make virtual visits based on their last visit for input by assuming the body weight is reduced in the near future. Since reducing body weight by a certain ratio is equivalent to reducing Body Mass Index (BMI) by the same ratio, and other physical examination factors like body fat and waist circumference might also be changed together with body weight, we determine which factors should be modified together alongside body weight (or BMI, equivalently) and modify them based on the following principles:

1. For BMI, modify it with the same ratio of body weight change.

2. For factors which are highly correlated with BMI (correlation coefficient of BMI and this factor is larger than 0.7), modify it according to the following equation:

$$y_{new} = s(BMI_{new} - BMI_{ori}) + y_{ori}, \tag{1}$$

where $BMI_{ori}$ and $BMI_{new}$ are respectively the original and the newly modified BMIs, $y_{ori}$ is the original value of the highly correlated factor, $s$ is the gender-dependent slope of the best fit line on the $xy$-plane using BMI as $x$ and the highly correlated factor as $y$, and $y_{new}$ is the newly modified version of the highly correlated factor. The correlation coefficient and $s$ are both calculated using only the training set.

3. For factors which are not highly correlated with BMI but can be derived by factors which are highly correlated with BMI, modify the latter factors based on the previous principle, and then use the newly modified version of the latter factors to derive the former factor.

The virtual visits are then appended to the original visits for input, and feed the new input to the trained models. If the new output probability is lower, the amount of body weight reduction of virtual visits for input could be used as the suggestion to visitors. Note that each virtual visit is assumed to appear one year after its previous visit, and to ensure the total number of visits for input is 5, the earliest visits are removed if needed.

## Experimental results and discussion

### Experimental setup

Due to the high memory consumption at the feature extraction stage, we extract features in a container running on a machine with Intel(R) Xeon(R) Gold 6154 CPU, occupying 4 CPU cores and 90 GB of host memory. On the other hand, the model training stage is conducted on a laptop with i7–8550U CPU and 20 GB of host memory.

For the hyperparameters of the models, the maximal number of estimators for XGBoost and LightGBM is set to 1,000, and the number of estimators which achieves the best validation loss in a fold is used for this fold. For random forest, a grid search is performed between 100 and 500 with a step size of 100, and between 600 and 1,000 with a step size of 50, to select the optimal number of estimators in each fold. This approach is necessary because the scikit-learn [15] implementation of random forest does not provide an interface for obtaining intermediate results of the training process. The maximal depths are respectively set to 5 for XGBoost and 15 for LightGBM and random forest, and the number of leaves for LightGBM is set to 20. The learning rates for XGBoost and LightGBM are empirically set to 0.01. Other hyperparameters, such as regularization terms, are left as default.

The main evaluation metrics are precision, recall, and F1-score, which are defined as follows:

$$precision = \frac{TP}{TP + FP}, \tag{2}$$

$$recall = \frac{TP}{TP + FN}, \text{ and} \tag{3}$$

$$F1 - score = \frac{2*precision*recall}{precision + recall}, \tag{4}$$

where $TP$, $FP$, and $FN$ are respectively true positive, false positive, and false negative.

### Results: Cross validation

Fig 7 shows the training and validation cross-entropy losses for the five folds of the three model types, and Table 3 shows the total training time in minutes and the average of the best validation cross-entropy losses across the five folds for each of the three model types. For XGBoost and LightGBM, the validation losses remain similar while the training losses trend downward when using more than 500 estimators, indicating that the training process does suffer from an over fitting problem. Besides, these two model types achieve similar average validation losses within acceptable training time. For random forest, the validation losses of all folds are obviously higher than those of XGBoost and LightGBM, and the training time is also longer. For the following experiments, we use XGBoost as the model type and the trained models which achieve lowest validation loss in each individual fold are used since the average of the best validation loss across the five folds is the lowest.

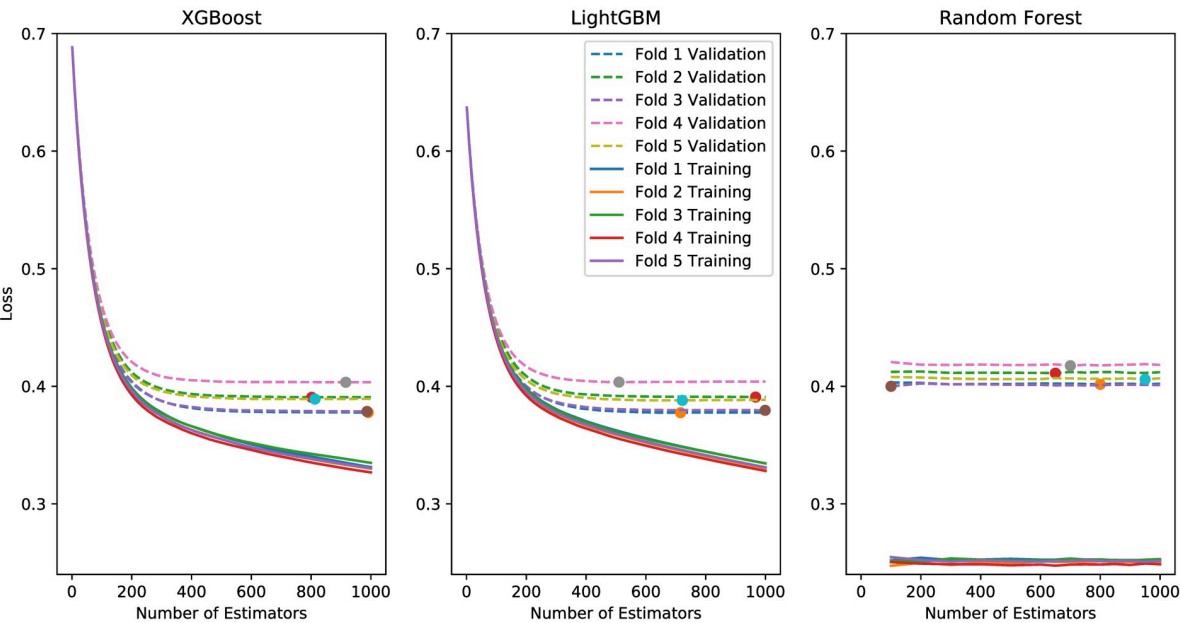

**Fig 7. Training and validation cross-entropy losses for the five folds of the three model types.** Dotted lines are validation losses and solid lines are training losses. Min values of each line for validation losses are marked with a solid circle.

## Results: Comparing with baseline method

The classification results of the baseline method and our models for the test set are respectively shown in Tables 4 and 5, where the baseline method is to use the positivity of the last visit for input to predict the output result. As shown in the tables, our models have slightly lower precision but much higher recall than the baseline method, thus leading to higher F1-score. The one-tailed t-test comparing the classification results of the test samples from the two methods also reveals that the results of our models are significantly closer to the ground truth, with a *p*-value of less than 0.001. We also show the Receiver Operating Characteristic (ROC) curves on

**Table 3. The total training time in minutes and the average of the best validation cross-entropy losses across the five folds for each of the three model types.**

| Model \ Metric | Total Training Time | Average validation Loss |
|---|---|---|
| XGBoost | 65 | 0.387860 |
| LightGBM | 5 | 0.387873 |
| Random Forest | 140 | 0.407213 |

**Table 4. Prediction results for the baseline method.** Precision, recall, and F1-score are respectively 71.98%, 69.43%, and 70.68%.

| Ground Truth \ Prediction | Negative | Positive |
|---|---|---|
| Negative | 42,849 (86.31%) | 6,799 (13.69%) |
| Positive | 7,690 (30.57%) | 17,464 (69.43%) |

**Table 5. Prediction results for our models.** Precision, recall, and F1-score are respectively 69.59%, 77.90%, and 73.51%.

| Ground Truth \ Prediction | Negative | Positive |
|---|---|---|
| Negative | 41,084 (82.75%) | 8,564 (17.25%) |
| Positive | 5,559 (22.10%) | 19,595 (77.90%) |

the test data for models from the five folds in Fig 8, where the curves of the 5 folds are very close to each other, indicating that the data splitting for the whole training set is balance. The lowest and the highest Area Under the ROC curves (AUROCs) are respectively 88.89% and 88.94%. We did also try neural network [8], but the performances of classification are not significantly different.

Precision, recall, and F1-score for different visit counts for our models are shown in Fig 9. As shown in the figure, values of precision increase with the number of visits. Values of recall and F1-score are also generally on the rise, except for those with four visits, which may lead to unstable results due to less data (4,286 people). Above results may suggest that more visits for a visitor leads to more accurate prediction results for hypertension. More accurate predictions enable earlier intervention when the prediction results are positive, and encouraging visitors to perform health checkups more often helps collect more comprehensive data, thereby refining the models.

Precision, recall, and F1-score for having hypertension or not in the past are shown in Fig 10. For the case of having hypertension in the past, our model has similar precision (baseline:

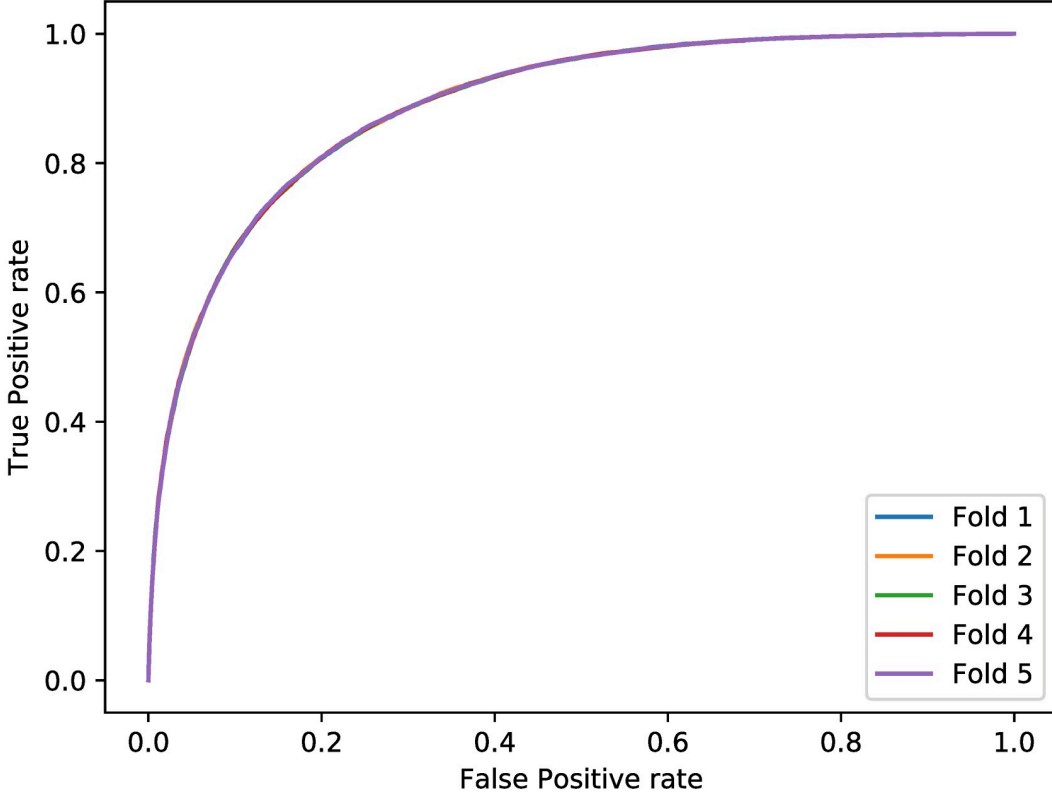

**Fig 8. ROC curves for each fold.**

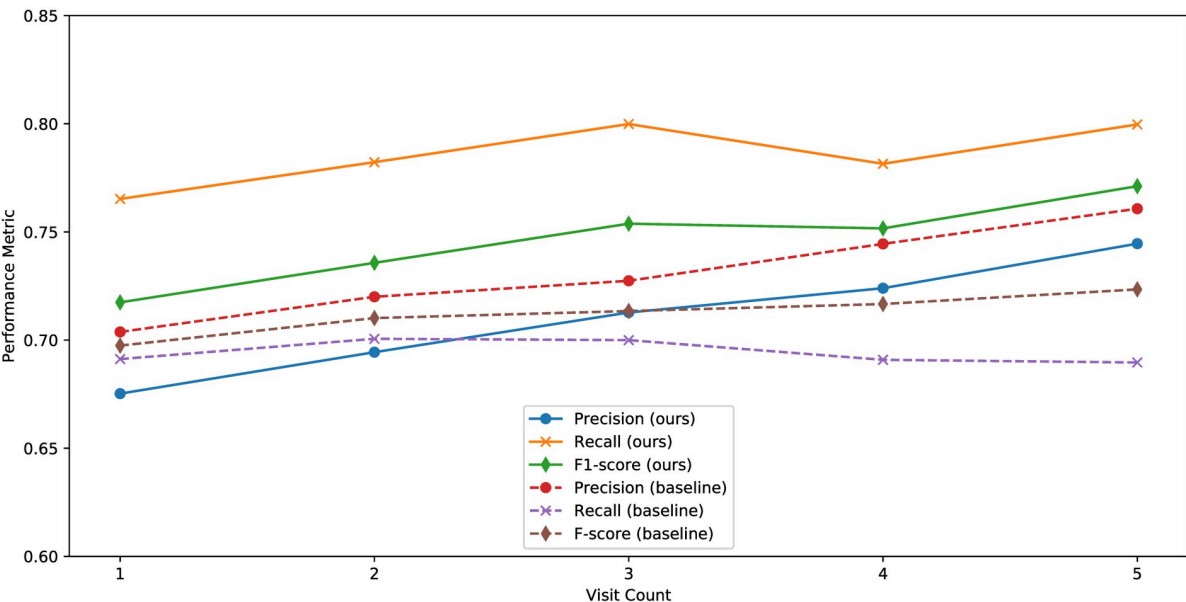

**Fig 9. Precision, recall, and F1-score for different visit counts.**

71.98%, ours: 72.61%) and higher recall (baseline: 89.16%, ours: 93.80%) than the baseline method, meaning that our models can tell whether a visitor having hypertension in the past can control their blood pressure in the future better than the baseline method. For the case of having no hypertension in the past, our models perform with precision of 42.76%, recall of 21.94%, and F1-score of 29.00%. But the baseline method, which uses only the historical positivity for prediction, cannot find anyone who will get hypertension in the future, thus the precision, recall and F1-score are all zero.

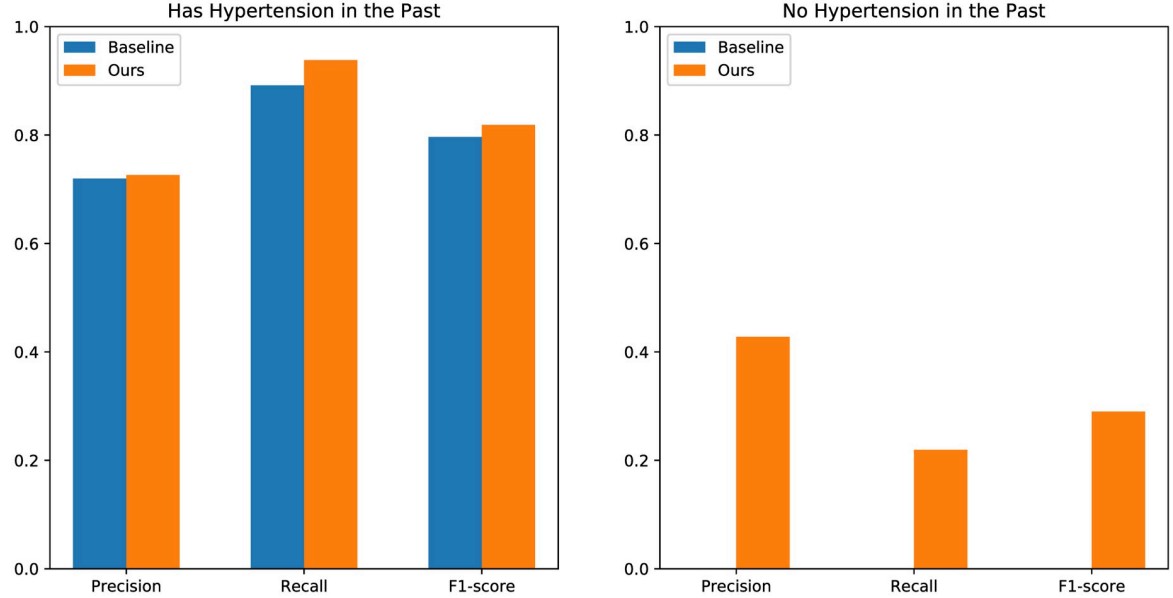

**Fig 10. Precision, recall, and F1-score for the cases of having hypertension or not in the past.**

## Results: Model performance using different factors

Because 1) not all health checkup items are helpful in predicting blood pressure, 2) different health checkup centers may have different examination items, and 3) using fewer features leads to faster training and inference time, selecting important factors not only saves computational resources but also makes the research more widely applicable. Therefore, we conducted experiments on top-20 important factors selected by our XGBoost models. For a single factor like body weight, since its' importance might be different in each fold and each visit, we sum its' importance for different visits in one fold, and average the importance for different folds as its' final importance. The top-20 important factors and their importance are listed in Table 6. While the importances of current blood pressure, age and body weight related factors are intuitive, the hearing are correlated with age (correlation coefficient: 0.39), and the importances of estimated glomerular filtration rate, hematocrit, vital capacity related, albumin globulin ratio, and white blood cell classification related factors have been discussed in [16–20].

The precisions, recalls, and F1-scores of models trained using only top-1 to top-20 important factors are illustrated in Fig 11, while the corresponding training time, precisions, recalls, and F1-scores for the top-5, top-10, and top-20 results are presented in Table 7. Despite the fluctuations in precisions and recalls as $n$ varies, the F1-scores remain relatively consistent when using 10 or more factors. Additionally, the training time of using only the top-$n$ factors is significantly less than that of using all the factors. Precisions, recalls, and F1-scores for having hypertension or not in the past are shown in Fig 12. When $n$ varies, the performances remain similar for the case of individuals with a history of hypertension, but they exhibit significant fluctuations for those with no hypertension in the past, especially when $n$ is small.

**Table 6. Top 20 important factors.**

| Rank | Feature Name | Unit | Importance | Remark |
|---|---|---|---|---|
| 1 | Mean arterial pressure | mmHg | 8.99% | SBP × 1/3 + DBP × 2/3 |
| 2 | Has hypertension | | 3.05% | If SBP ≥ 130 or DBP ≥ 80 |
| 3 | SBP | mmHg | 1.26% | |
| 4 | DBP | mmHg | 1.15% | |
| 5 | Estimated glomerular filtration rate | | 0.87% | |
| 6 | Is liver ultrasonography normal | | 0.85% | 0: normal, 1: abnormal |
| 7 | Forced expiratory volume in 1 sec | L | 0.85% | |
| 8 | Age | year | 0.82% | |
| 9 | BMI | $kg/m^2$ | 0.82% | Weight / square of height (in meter) |
| 10 | Pulse pressure difference | mmHg | 0.81% | |
| 11 | Hematocrit | % | 0.79% | |
| 12 | Forced vital capacity | L | 0.78% | |
| 13 | Maximum mid-expiratory flow | L/sec | 0.78% | |
| 14 | Waist circumference | cm | 0.78% | |
| 15 | WBC classification: lymphocytes | % | 0.77% | |
| 16 | Albumin globulin ratio | | 0.77% | Albumin / globulin |
| 17 | Waist-hip ratio | | 0.77% | Waist circumference / hip circumference |
| 18 | WBC classification: eosinophil | % | 0.77% | |
| 19 | Albumin | g/dL | 0.74% | |
| 20 | Right ear hearing | dB | 0.73% | |

SBP: systolic blood pressure. DBP: diastolic blood pressure. WBC: white blood cell.

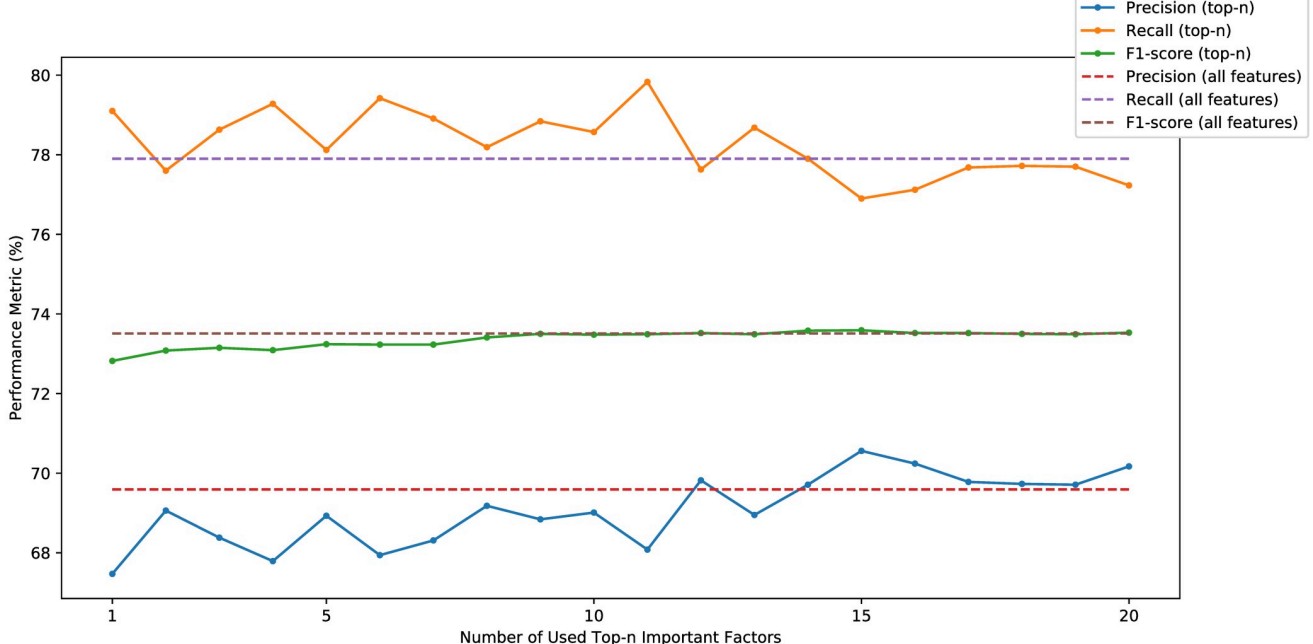

**Fig 11. Prediction results using top-n important factors.**

These results may suggest that the selection of factors is important for predicting future hypertension for the case of having no hypertension in the past.

We also train models using factors of the Framingham risk score, which uses age, gender, total cholesterol, high density lipoprotein cholesterol, smoking habits, systolic blood pressure, and glucose to predict long term risk of cardiovascular disease or hypertension [21–23]. The evaluation result is shown in Table 8. Both precision and recall are lower than our models using top-5, top-10, and top-20 important factors, and the precision and F1-score are even lower than the baseline method. A one-tailed t-test comparing the classification results of the test samples using factors from the Framingham risk score and our top-20 important factors shows that the latter are significantly closer to the ground truth, with a *p*-value of less than 0.001.

## Results: Use of models for hypertension prevention

By examining the training set, we find that waist circumference, hip circumference, and body fat ratio are highly correlated with BMI. For waist-hip ratio, even though it is not highly correlated with BMI, it can be derived from waist circumference and hip circumference. The scatter plots of waist circumference, hip circumference, waist-hip ratio, and body fat ratio versus BMI are shown in Fig 13. We have also calculated the relationship between BMI and other factors

**Table 7. Training time (in minute), precision, recall, and f1-score for classification results using the top-5, top-10, and top-20 important factors.**

| *n* \ Metric | Training Time | Precision | Recall | F1-score |
|:---:|:---:|:---:|:---:|:---:|
| **5** | 4 | 68.93% | 78.12% | 73.24% |
| **10** | 7 | 69.01% | 78.57% | 73.48% |
| **20** | 8 | 70.71% | 77.23% | 73.53% |

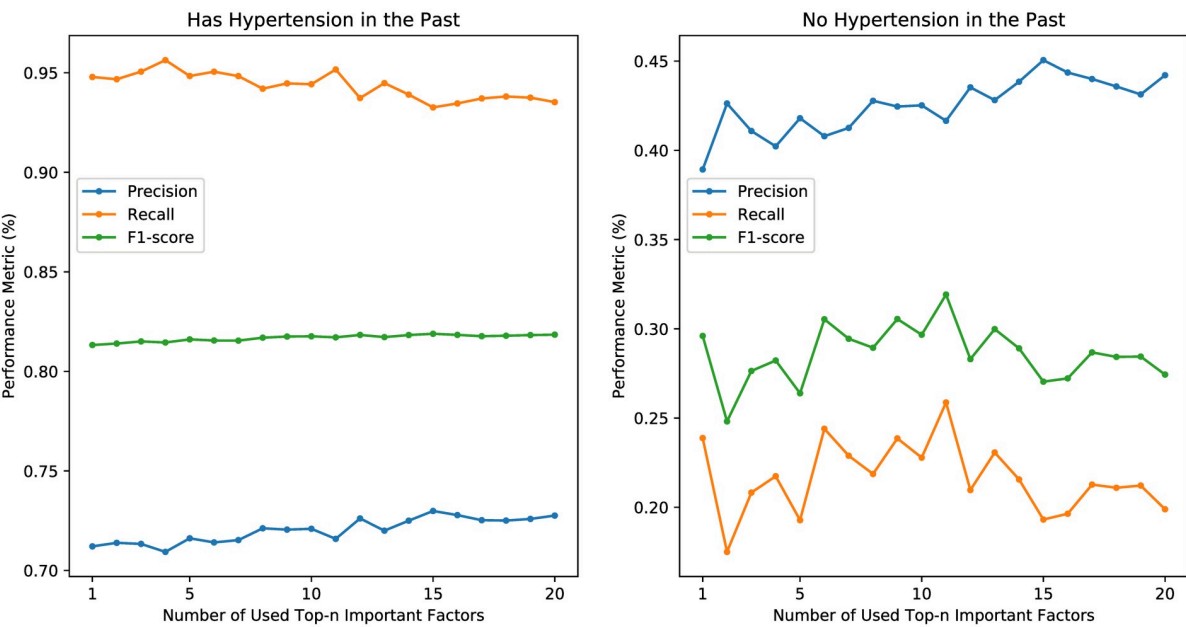

**Fig 12. Prediction results using top-n important factors for the cases of having hypertension in the past or not.**

in the top-20 important list, but all other factors are not highly correlated with BMI, or could not be derived from factors that are highly correlated with BMI. Therefore, when making virtual visits for input, the factors being modified are body weight, BMI, waist circumference, hip circumference, body fat ratio, and waist-hip ratio.

To observe whether the change of model judgment before and after adding virtual visits can be used as suggestions for a visitor who are predicted to have hypertension in the future (i.e. true positive and false positive cases, since true negative and false negative cases are not considered because the classification results suggest that they may not require further action for blood pressure control.), we make virtual visits based on their last visit for model input by reducing the BMI of the last visit to 24 if that BMI is larger than 24. Different numbers of virtual visits are added and the observed change of model judgment is shown in Fig 14. For the 19,427 true positive cases classified by the models trained using the top-20 important factors, 12,58 individuals have a BMI larger than or equal to 24. After incorporating 1 to 5 virtual visits, a substantial proportion of these individuals, specifically 5.48% to 6.70%, were reclassified as negative, potentially leading to a reduction in the dosage of medication. For the 8.257 false positive cases, 5,009 individuals have a BMI larger than or equal to 24, and 18.91% to 20.94% of the 5,009 individuals were reclassified as negative after incorporating 1 to 5 virtual visits. Despite the higher number of affected individuals in the false positive group, our approach still may by useful since weight control is beneficial for health [24, 25], even if blood pressure is normal. To validate the effectiveness of these findings in the absence of real data, we followed

**Table 8. Prediction results for our models using factors of Framingham risk score as features.** Precision, recall, and F1-score are respectively 67.69%, 73.99%, and 70.70%.

| Ground Truth \ Prediction | Negative | Positive |
|---|---|---|
| Negative | 40,764 (82.11%) | 8,884 (17.89%) |
| Positive | 6,543 (26.01%) | 18,611 (73.99%) |

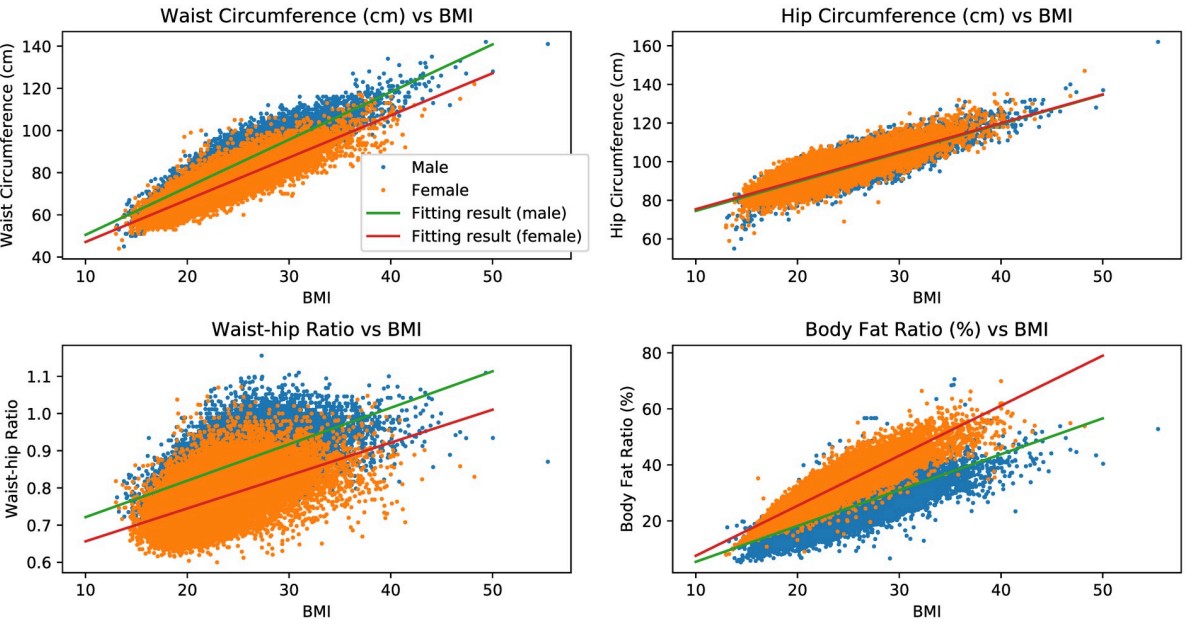

**Fig 13. Waist circumference, hip circumference, waist-hip ratio, and body fat ratio versus BMI.**

the validation methodology of previous studies by Chi et al. [26] and Dogan et al. [27]. The one-tailed t-test results demonstrate that, compared to the raw classification results from models trained using the top-20 important factors, incorporating virtual visits significantly reduces the number of people predicted to have hypertension in the future ($p < 0.001$), while maintaining a high correlation between the predicted results before and after incorporating virtual

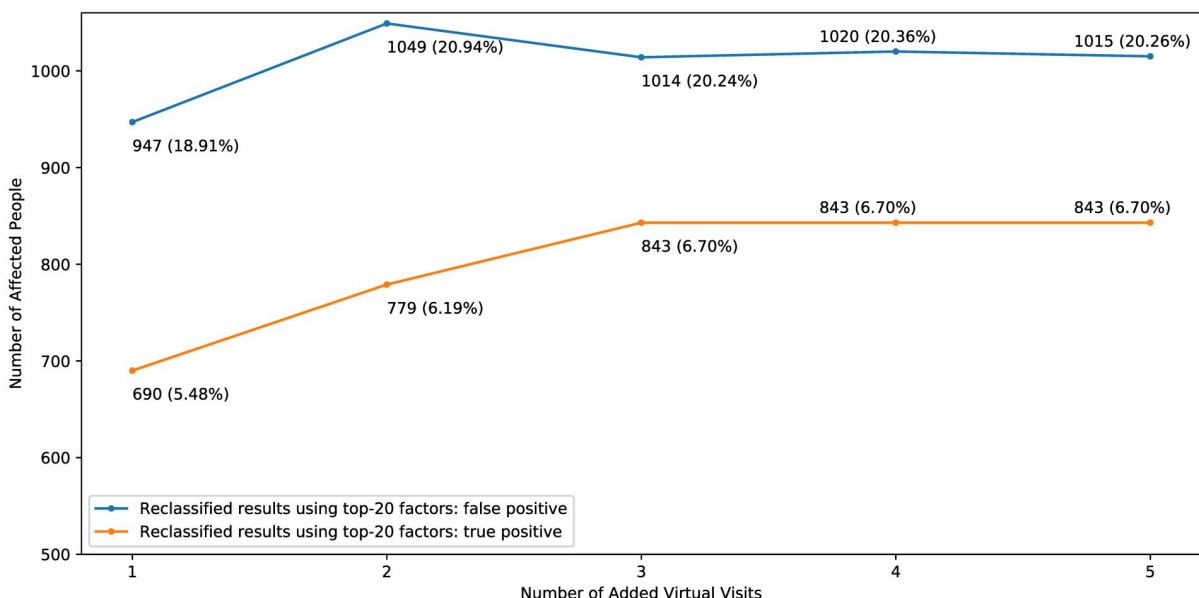

**Fig 14. Number of people (out of the 12,585 or 5,009 people who are originally respectively true positive or false positive cases and are with BMI $\geq$ 24) who are judged as negative after adding different numbers of virtual visits.**

visits (Matthews correlation coefficient >0.9). A rising trend of numbers of affected people is also observed when adding more virtual visits, which may indicate that if one can control their body weight for a longer period, they will be with lower probability of having hypertension in the future. Despite the fact that the numbers of people who are judged as negative after adding virtual visits are not very large, the above results show our ability to suggest visitors for weight control.

## Conclusions and future work

In this paper, we use machine learning models to predict the risk of hypertension for one's next visit to a health checkup center using their routine health checkup data in the past with 69.59% of precision, 77.90% of recall, and 73.51% of F1-score, which outperform the baseline system using only the positivity of one's last visit in the past to predict the risk of hypertension for their next visit. Experimental results also suggest that if a visitor visits more times, then the prediction result is more accurate, thereby enabling earlier intervention and encouraging visitors to perform health checkups more often. We also conducted experiments using different factors, and our selected top-n important factors achieved more accurate results than those obtained using the factors of the Framingham risk score. Finally, we examine the ability of suggesting visitors for weight control by using our model. By incorporating virtual visits and appending them to the original model input, where the virtual visits assume that one's body weight can be reduced in the near future, our model reevaluates approximately 5.48% to 6.70% of true positive cases with high BMI, reclassifying them as negative. This reevaluation potentially leads to a reduction in the dosage of medication. Besides, when adding more virtual visits, the ratios of people who are rejudged to negative show a rising trend, which may indicate that controlling body weight for a longer time helps to reduce the risk of having hypertension in the future. In the future, this work may be improved by adding features extracted from images of electrocardiography or fundus photography. Since the sizes of these two types of images are both large, extracting proper embeddings or features from the raw images to train models at a lower cost will be one of the challenges.

## Supporting information

**S1 Appendix. The 266 features for one visit (listed according to the practical implementation order of dimensions).** For convenience, we categorize the features into several groups, with the dimensionality denoted after each group name. The actual range of values for each feature depends on the type of physical examination or the design of the questionnaire. (PDF)

## Acknowledgments

We thank the National Center for High-performance Computing (NCHC) for providing computational and storage resources. All or part of the data used in this research were authorized by, and received from MJ Health Research Foundation (Authorization Code: MJHRF21019015C). Any interpretation or conclusion described in this paper does not represent the views of MJ Health Research Foundation.

## Author Contributions

**Conceptualization:** Ta-Wei Chu.

**Data curation:** Ta-Wei Chu.

**Formal analysis:** Chung-Che Wang.

**Investigation:** Chung-Che Wang.

**Methodology:** Chung-Che Wang.

**Project administration:** Ta-Wei Chu, Jyh-Shing Roger Jang.

**Resources:** Ta-Wei Chu, Jyh-Shing Roger Jang.

**Software:** Chung-Che Wang.

**Supervision:** Ta-Wei Chu, Jyh-Shing Roger Jang.

**Validation:** Ta-Wei Chu, Jyh-Shing Roger Jang.

**Visualization:** Chung-Che Wang.

**Writing – original draft:** Chung-Che Wang.

**Writing – review & editing:** Chung-Che Wang.

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
