## [Decision Letter · Decision Letter 0]

24 Jun 2024

PONE-D-23-27067Next-visit prediction and prevention of hypertension using large-scale routine health checkup dataPLOS ONE

Dear Dr. Wang,

Thank you for submitting your manuscript to PLOS ONE. After careful consideration, we feel that it has merit but does not fully meet PLOS ONE’s publication criteria as it currently stands. Therefore, we invite you to submit a revised version of the manuscript that addresses the points raised during the review process.

We look forward to receiving your revised manuscript.

Kind regards,

Sana Sadiq Sheikh

Academic Editor

PLOS ONE

Journal Requirements:

4. We noted in your submission details that a portion of your manuscript may have been presented or published elsewhere. "No. Despite that this paper is an extension of our previous work presented the 2022 APAMI conference, but the conference did not publish proceeding."

**Additional Editor Comments:**

Reviewer 1:

Your study on predicting hypertension risk using XGBoost models and simulating weight reduction's impact on health indicators for hypertension prevention is commendable for its innovative approach. However, to further strengthen the credibility and applicability of your findings, I suggest considering the following:

1. Elaborate on the methodology used to determine correlations among health indicators and the rationale behind modifying specific factors during the creation of virtual visits. This will enhance the understanding of your approach.

2. Provide additional details on the statistical tests employed for validation and discuss any potential limitations in the validation process. This will ensure a more robust evaluation of the effectiveness of virtual visits.

3. Contextualize the observed changes in reclassifying individuals with high BMI after incorporating virtual visits within the dataset's overall size and clinical significance. This will help understand the practical implications of these changes for hypertension prevention.

4. Consider comparing your XGBoost models against other state-of-the-art models or risk prediction approaches. This comparative analysis would provide a broader perspective on the model's effectiveness.

5. Discuss in more detail the practical implications of your findings in clinical or preventive healthcare settings. Additionally, expand on the challenges and limitations associated with integrating features from medical imaging for further model enhancement.

6. Consider relocating legends within the graphical presentation to ensure they do not obstruct any part of the graph lines or bars. This adjustment will improve visual clarity.

7. The reasoning behind including the people count on the scale is unclear, especially considering the lack of explanation regarding the results in Figure 8 in relation to this count. Please either elucidate the results with reference to the people count or consider removing the scale and people count line for clarity.

Your study has the potential to significantly contribute to preventive healthcare strategies. Strengthening these aspects will further enhance the credibility and applicability of your research findings. Your valuable contributions to this field are appreciated, and the anticipation of advancements in your work is eagerly awaited.

Reviewer 2:

This paper uses XGBoost models to predict the occurrence of hypertension at the next patient visit. The authors employ a well-known, standard dataset. They also suggest methods of blood pressure controlling to patients with high future probability of hypertension.

Pros of the paper:

The method of training set and test construction seems justified.

The method of adding virtual visits for weight control is innovative

Cons of the paper:

The authors have not motivated the use of XGBoost models. It is true that XGBoost has become a sort of standard for machine learning application researchers. However, it has serious disadvantages as well (e.g., overfitting). The authors needed to thoroughly justify the use of XGBoost for the problem of hypertension prediction.

And the problem statement itself is also not clear to me. If the problem statement is to solve the problem of hypertension prediction, then the authors needed to try out multiple ML algorithms over multiple related datasets to justify the applicability of a single algorithm. If the problem is to solve the given dataset, then why only XGBoost? The very sudden reference to XGBoost in the first sentence of the Abstract is in fact, confusing. To me, it shows a lack of clarity as to the method of contributing to the research literature. Applying XGBoost to a problem cannot be the motivation for a paper - an algorithm is a means to achieve an end. It itself cannot be labeled as research - unless there is a strong need to do so (which is not present anywhere in the paper).

The gap analysis shown in Table 1 is weak. Summing up, the major gap is the increased length of the dataset across several dimensions. One would expect the mention of different algorithms/approaches. The second-last column seems unnecessary since selection of more features is not a concrete research gap data.

The authors have not mentioned all 266 features in the appendix

The authors have not mentioned the pros and cons of feature vector concatenation

The authors have used mean to replace missing values - this is the least preferred method unless the number of missing values to fill-in are extremely small (less than 0.5% for example). Better methods include interpolation and chained equations.

The authors have not justified the use of 5 folds.

There is no concrete machine learning methodology diagram in the whole paper.

The justification of the authors that different factors (features) do affect the final result is a very basic fact of machine learning community. This adds no value to the paper.

The ROC curve of all five folds is exactly the same - which raises doubts that a potential overfitting scenario has occurred, or the number of folds needs to increase.

Overall, a more concrete machine learning methodology needs to be applied with proper motivations and justifications.

Reviewers' comments:

Reviewer's Responses to Questions

**Comments to the Author**

1. Is the manuscript technically sound, and do the data support the conclusions?

Reviewer #1: Partly

Reviewer #2: Partly

2. Has the statistical analysis been performed appropriately and rigorously? 

Reviewer #1: Yes

Reviewer #2: Yes

3. Have the authors made all data underlying the findings in their manuscript fully available?

Reviewer #1: Yes

Reviewer #2: Yes

4. Is the manuscript presented in an intelligible fashion and written in standard English?

Reviewer #1: Yes

Reviewer #2: Yes

5. Review Comments to the Author

Reviewer #1: Your study on predicting hypertension risk using XGBoost models and simulating weight reduction's impact on health indicators for hypertension prevention is commendable for its innovative approach. However, to further strengthen the credibility and applicability of your findings, I suggest considering the following:

1. Elaborate on the methodology used to determine correlations among health indicators and the rationale behind modifying specific factors during the creation of virtual visits. This will enhance the understanding of your approach.

2. Provide additional details on the statistical tests employed for validation and discuss any potential limitations in the validation process. This will ensure a more robust evaluation of the effectiveness of virtual visits.

3. Contextualize the observed changes in reclassifying individuals with high BMI after incorporating virtual visits within the dataset's overall size and clinical significance. This will help understand the practical implications of these changes for hypertension prevention.

4. Consider comparing your XGBoost models against other state-of-the-art models or risk prediction approaches. This comparative analysis would provide a broader perspective on the model's effectiveness.

5. Discuss in more detail the practical implications of your findings in clinical or preventive healthcare settings. Additionally, expand on the challenges and limitations associated with integrating features from medical imaging for further model enhancement.

6. Consider relocating legends within the graphical presentation to ensure they do not obstruct any part of the graph lines or bars. This adjustment will improve visual clarity.

7. The reasoning behind including the people count on the scale is unclear, especially considering the lack of explanation regarding the results in Figure 8 in relation to this count. Please either elucidate the results with reference to the people count or consider removing the scale and people count line for clarity.

Your study has the potential to significantly contribute to preventive healthcare strategies. Strengthening these aspects will further enhance the credibility and applicability of your research findings. Your valuable contributions to this field are appreciated, and the anticipation of advancements in your work is eagerly awaited.

Reviewer #2: This paper uses XGBoost models to predict the occurrence of hypertension at the next patient visit. The authors employ a well-known, standard dataset. They also suggest methods of blood pressure controlling to patients with high future probability of hypertension.

Decision: Reject

Pros of the paper:

The method of training set and test construction seems justified.

The method of adding virtual visits for weight control is innovative

Cons of the paper:

The authors have not motivated the use of XGBoost models. It is true that XGBoost has become a sort of standard for machine learning application researchers. However, it has serious disadvantages as well (e.g., overfitting). The authors needed to thoroughly justify the use of XGBoost for the problem of hypertension prediction.

And the problem statement itself is also not clear to me. If the problem statement is to solve the problem of hypertension prediction, then the authors needed to try out multiple ML algorithms over multiple related datasets to justify the applicability of a single algorithm. If the problem is to solve the given dataset, then why only XGBoost? The very sudden reference to XGBoost in the first sentence of the Abstract is in fact, confusing. To me, it shows a lack of clarity as to the method of contributing to the research literature. Applying XGBoost to a problem cannot be the motivation for a paper - an algorithm is a means to achieve an end. It itself cannot be labeled as research - unless there is a strong need to do so (which is not present anywhere in the paper).

The gap analysis shown in Table 1 is weak. Summing up, the major gap is the increased length of the dataset across several dimensions. One would expect the mention of different algorithms/approaches. The second-last column seems unnecessary since selection of more features is not a concrete research gap data.

The authors have not mentioned all 266 features in the appendix

The authors have not mentioned the pros and cons of feature vector concatenation

The authors have used mean to replace missing values - this is the least preferred method unless the number of missing values to fill-in are extremely small (less than 0.5% for example). Better methods include interpolation and chained equations.

The authors have not justified the use of 5 folds.

There is no concrete machine learning methodology diagram in the whole paper.

The justification of the authors that different factors (features) do affect the final result is a very basic fact of machine learning community. This adds no value to the paper.

The ROC curve of all five folds is exactly the same - which raises doubts that a potential overfitting scenario has occurred, or the number of folds needs to increase.

Overall, a more concrete machine learning methodology needs to be applied with proper motivations and justifications.

6. PLOS authors have the option to publish the peer review history of their article (what does this mean?). If published, this will include your full peer review and any attached files.

Reviewer #1: **Yes: **Muhammad Aasim

Reviewer #2: No

---

## [Author Response · Author response to Decision Letter 0]

9 Aug 2024

Journal Requirements: Please ensure that your manuscript meets PLOS ONE's style requirements, including those for file naming.

Response: We have updated the manuscript in accordance with the style guidelines as much as possible.

Journal Requirements: Please note that PLOS ONE has specific guidelines on code sharing for submissions in which author-generated code underpins the findings in the manuscript. In these cases, all author-generated code must be made available without restrictions upon publication of the work.

Response: We have published our source code to GitHub, which is available at: https://github.com/geniusturtle6174/next-visit-prediction-and-prevention-of-hypertension, and the DOI is: 10.5281/zenodo.13162787

Journal Requirements: Consider depositing your raw data in a repository to ensure your work is read, appreciated and cited by the largest possible audience.

Response: Due to the data provider's policy, we regret that we are unable to deposit the data.

Journal Requirements: We noted in your submission details that a portion of your manuscript may have been presented or published elsewhere. "No. Despite that this paper is an extension of our previous work presented the 2022 APAMI conference, but the conference did not publish proceeding." Please clarify whether this [conference proceeding or publication] was peer-reviewed and formally published. If this work was previously peer-reviewed and published, in the cover letter please provide the reason that this work does not constitute dual publication and should be included in the current manuscript.

Response: The work presented at the 2022 APAMI conference was peer-reviewed, but the full text has not been published. Therefore, the current work does not constitute dual publication. We have included emails from the 2022 APAMI conference: one is the acceptance letter with review comments (forwarded from the corresponding author), and the other is the official statement confirming that only the abstract will be published.

Journal Requirements: Please include your full ethics statement in the ‘Methods’ section of your manuscript file. In your statement, please include the full name of the IRB or ethics committee who approved or waived your study, as well as whether or not you obtained informed written or verbal consent. If consent was waived for your study, please include this information in your statement as well.

Response: We have updated the manuscript (“Dataset” subsection of the “Materials and methods” section) accordingly.

Comments (Reviewer #1): Elaborate on the methodology used to determine correlations among health indicators and the rationale behind modifying specific factors during the creation of virtual visits. This will enhance the understanding of your approach.

Response: The methodology used to determine correlations is correlation coefficient calculated using only the training set, and the rationale behind modifying specific factors during the creation of virtual visits is those specific factors are are highly correlated with BMI (or can be derived by factors which are highly correlated with BMI). We have updated the manuscript (“Use of Models for Hypertension Prevention” subsection of the “Materials and methods ” section) accordingly.

Comments (Reviewer #1): Provide additional details on the statistical tests employed for validation and discuss any potential limitations in the validation process. This will ensure a more robust evaluation of the effectiveness of virtual visits.

Response: The comparison is performed between the raw and the reclassified results using the models trained using the top-20 important factors, and one of the limitations is the absence of real data. We have updated the manuscript (“Results: Use of Models for Hypertension Prevention” subsection of the “Experimental Results and Discussion” section) accordingly.

Comments (Reviewer #1): Contextualize the observed changes in reclassifying individuals with high BMI after incorporating virtual visits within the dataset's overall size and clinical significance. This will help understand the practical implications of these changes for hypertension prevention.

Response: We have added experimental results and discussions for false positive cases in the manuscript (“Results: Use of Models for Hypertension Prevention” subsection of the “Experimental Results and Discussion” section). True negative and false negative cases are not considered because the classification results suggest that they may not require further action for blood pressure control.

Comments (Reviewer #1): Consider comparing your XGBoost models against other state-of-the-art models or risk prediction approaches. This comparative analysis would provide a broader perspective on the model's effectiveness.

Comments (Reviewer #2): The authors have not motivated the use of XGBoost models. It is true that XGBoost has become a sort of standard for machine learning application researchers. However, it has serious disadvantages as well (e.g., overfitting). The authors needed to thoroughly justify the use of XGBoost for the problem of hypertension prediction.

Comments (Reviewer #2): If the problem statement is to solve the problem of hypertension prediction, then the authors needed to try out multiple ML algorithms over multiple related datasets to justify the applicability of a single algorithm. If the problem is to solve the given dataset, then why only XGBoost? The very sudden reference to XGBoost in the first sentence of the Abstract is in fact, confusing. To me, it shows a lack of clarity as to the method of contributing to the research literature. Applying XGBoost to a problem cannot be the motivation for a paper - an algorithm is a means to achieve an end. It itself cannot be labeled as research - unless there is a strong need to do so (which is not present anywhere in the paper).

Comments (Reviewer #2): Overall, a more concrete machine learning methodology needs to be applied with proper motivations and justifications.

Response: We have added the results of random forest and LightGBM, as XGBoost, random forest, and LightGBM have demonstrated the best performance in various previous studies. Additionally, a subsection on the results of cross validation has been added to clarify that overfitting has not occurred.

Comments (Reviewer #1): Discuss in more detail the practical implications of your findings in clinical or preventive healthcare settings. Additionally, expand on the challenges and limitations associated with integrating features from medical imaging for further model enhancement.

Response: The implication of “if a visitor visits more times, then the prediction result is more accurate” is that it enables earlier intervention when the prediction results are positive and encourages visitors to perform health checkups more frequently. The implication of “reclassifying true positive cases with high BMI as negative using virtual visits” is that it could potentially lead to a reduction in the dosage of medication. One of the challenges of integrating features from medical imaging is determining how to extract proper embeddings or features from the raw images to train models at a lower cost. We have updated the manuscript (“Results: Comparing with Baseline Method” and “Results: Use of Models for Hypertension Prevention” subsection of the “Experimental Results and Discussion” section, and the “Conclusions and Future Work” section) accordingly.

Comments (Reviewer #1): Consider relocating legends within the graphical presentation to ensure they do not obstruct any part of the graph lines or bars. This adjustment will improve visual clarity.

Response: We have relocated the legends of figure 10 and 11 (new figure index).

Comments (Reviewer #1): The reasoning behind including the people count on the scale is unclear, especially considering the lack of explanation regarding the results in Figure 8 in relation to this count. Please either elucidate the results with reference to the people count or consider removing the scale and people count line for clarity.

Response: We have removed the people counts in the titles of figure 10 (new figure index) and the curve for people count in figure 9 (new figure index).

Comments (Reviewer #2): The gap analysis shown in Table 1 is weak. Summing up, the major gap is the increased length of the dataset across several dimensions. One would expect the mention of different algorithms/approaches. The second-last column seems unnecessary since selection of more features is not a concrete research gap data.

Response: We have removed the second-last column (“Features that may not be in routine health checkup”) of table 1, and added a column of “Methods” for table 1.

Comments (Reviewer #2): The authors have not mentioned all 266 features in the appendix

Response: We have provided the list of all the 266 features in the appendix.

Comments (Reviewer #2): The authors have not mentioned the pros and cons of feature vector concatenation

Response: The advantage is that data from visitors with different visit counts can be utilized together during training, leading to more effective data usage. However, the disadvantage is that the concatenation process requires a bit of extra storage and computational resources when training models. We have updated the manuscript (“Feature Extraction and Output Definition” subsection of the “Materials and methods” section) accordingly.

Comments (Reviewer #2): The authors have used mean to replace missing values - this is the least preferred method unless the number of missing values to fill-in are extremely small (less than 0.5% for example). Better methods include interpolation and chained equations.

Response: We have changed to use linear interpolation and updated all the experimental results in the manuscript accordingly.

Comments (Reviewer #2): The authors have not justified the use of 5 folds.

Response: The use of 5 folds is based on previous work. We have updated the manuscript (last paragraph of introduction) accordingly.

Comments (Reviewer #2): There is no concrete machine learning methodology diagram in the whole paper.

Response: We have added an overview diagram of our methodology at the beginning of the “Materials and methods” section, and included a brief description of the machine learning algorithms in the “Train the Models” subsection of the same section.

Comments (Reviewer #2): The justification of the authors that different factors (features) do affect the final result is a very basic fact of machine learning community. This adds no value to the paper.

Response: Feature selection is basic in machine learning but beneficial for this paper due to 1) not all health checkup items are helpful in predicting blood pressure, 2) different health checkup centers may have different examination items, and 3) using fewer features leads to faster training and inference time. Therefore, selecting important features saves computational resources and makes the research more widely applicable. We have updated the manuscript (“Results: Model Performance Using Different Factors” subsection of the “Experimental Results and Discussion” section) accordingly.

Comments (Reviewer #2): The ROC curve of all five folds is exactly the same - which raises doubts that a potential overfitting scenario has occurred, or the number of folds needs to increase.

Response: Similar ROC curves indicate that the data splitting is balanced in our case. We have updated the manuscript (“Results: Comparing with Baseline Method” subsection of the “Experimental Results and Discussion” section) accordingly. Additionally, a subsection (“Results: Cross Validation”) detailing the results of cross-validation, including training and validation losses, has been added to clarify that overfitting has not occurred.

---

## [Decision Letter · Decision Letter 1]

11 Sep 2024

PONE-D-23-27067R1Next-visit prediction and prevention of hypertension using large-scale routine health checkup dataPLOS ONE

Dear Dr. Wang,

Thank you for submitting your manuscript to PLOS ONE. After careful consideration, we feel that it has merit but does not fully meet PLOS ONE’s publication criteria as it currently stands. Therefore, we invite you to submit a revised version of the manuscript that addresses the points raised during the review process.

Your manuscript has been evaluated by both of the previous reviewers, and their comments are appended below. Both reviewers are satisfied with your revisions, and they overall recommend publication. Before we proceed, Reviewer 1 has noted that the scales used in Fig. 8 are not consistent; please revise the x-axis scale for the Random Forest subfigure. We do not anticipate that further review will be necessary.

We look forward to receiving your revised manuscript.

Kind regards,

Hugh Cowley

Staff Editor

PLOS ONE

Journal Requirements:

Reviewers' comments:

Reviewer's Responses to Questions

**Comments to the Author**

1. If the authors have adequately addressed your comments raised in a previous round of review and you feel that this manuscript is now acceptable for publication, you may indicate that here to bypass the “Comments to the Author” section, enter your conflict of interest statement in the “Confidential to Editor” section, and submit your "Accept" recommendation.

Reviewer #1: All comments have been addressed

Reviewer #2: All comments have been addressed

2. Is the manuscript technically sound, and do the data support the conclusions?

Reviewer #1: Yes

Reviewer #2: Yes

3. Has the statistical analysis been performed appropriately and rigorously? 

Reviewer #1: Yes

Reviewer #2: No

4. Have the authors made all data underlying the findings in their manuscript fully available?

Reviewer #1: Yes

Reviewer #2: Yes

5. Is the manuscript presented in an intelligible fashion and written in standard English?

Reviewer #1: Yes

Reviewer #2: Yes

6. Review Comments to the Author

Reviewer #1: I am satisfied with the authors' responses, and I believe the manuscript has been significantly improved. However, I have one remaining concern regarding Figure 8. In the figure, both XGBoost and LightGBM are plotted on a scale that starts from 0 and ends at 1000, while the scale for Random Forest begins at 600 and ends at 1000. This discrepancy makes it difficult to compare Random Forest with the other two models. I recommend adjusting the scales to be consistent across all three models to facilitate a more accurate comparison. Subject to this change, I recommend the acceptance of the manuscript.

Reviewer #2: The authors have catered for all of my comments. They didn't respond to all, but the changes are there.

7. PLOS authors have the option to publish the peer review history of their article (what does this mean?). If published, this will include your full peer review and any attached files.

Reviewer #1: **Yes: **Muhammad Aasim

Reviewer #2: No

---

## [Author Response · Author response to Decision Letter 1]

27 Sep 2024

Journal Requirements: Please review your reference list to ensure that it is complete and correct. If you have cited papers that have been retracted, please include the rationale for doing so in the manuscript text, or remove these references and replace them with relevant current references. Any changes to the reference list should be mentioned in the rebuttal letter that accompanies your revised manuscript. If you need to cite a retracted article, indicate the article’s retracted status in the References list and also include a citation and full reference for the retraction notice.

Response: After carefully reviewing our reference list, we confirm that none of the cited papers or materials have been retracted.

Comments (Reviewer #1): I have one remaining concern regarding Figure 8. In the figure, both XGBoost and LightGBM are plotted on a scale that starts from 0 and ends at 1000, while the scale for Random Forest begins at 600 and ends at 1000. This discrepancy makes it difficult to compare Random Forest with the other two models. I recommend adjusting the scales to be consistent across all three models to facilitate a more accurate comparison. Subject to this change, I recommend the acceptance of the manuscript.

Response: We have adjusted the scale of the X-axis of the third subplot to [0, 1000]. Additionally, to ensure that the experimental results for Random Forest (RF) are not missing in the [0, 500] region, we conducted the corresponding experiments, and updated the content in the “Results: Cross Validation” and the “Experimental Setup” subsections of the “Experimental Results and Discussion” section, along with table 3. We have also found that we mistakenly named the files for Figures 7 and 8, and we have corrected the file names in the revised version.

Comments (Reviewer #2): The authors have catered for all of my comments. They didn't respond to all, but the changes are there.

Response: We have provided additional details in the “Results: Comparing with Baseline Method,” “Results: Model Performance Using Different Factors,” and “Results: Use of Models for Hypertension Prevention” subsections within the “Experimental Results and Discussion” section.

---

## [Editor Report · Decision Letter 2]

29 Oct 2024

Next-visit prediction and prevention of hypertension using large-scale routine health checkup data

PONE-D-23-27067R2

Dear Dr. Wang,

We’re pleased to inform you that your manuscript has been judged scientifically suitable for publication and will be formally accepted for publication once it meets all outstanding technical requirements.

Kind regards,

Hidetaka Hamasaki

Academic Editor

PLOS ONE
---

## [Editor Report · Acceptance letter]

4 Nov 2024

PONE-D-23-27067R2 

PLOS ONE

Dear Dr. Wang, 

I'm pleased to inform you that your manuscript has been deemed suitable for publication in PLOS ONE. Congratulations! Your manuscript is now being handed over to our production team.

Kind regards, 

on behalf of

Dr. Hidetaka Hamasaki 

Academic Editor

PLOS ONE